# Adhesive-Ceramic Interface Behavior in Dental Restorations. FEM Study and SEM Investigation

**DOI:** 10.3390/ma14175048

**Published:** 2021-09-03

**Authors:** Otilia Chirca, Cornelia Biclesanu, Anamaria Florescu, Dan Ioan Stoia, Anna Maria Pangica, Alexandru Burcea, Marius Vasilescu, Iulian Vasile Antoniac

**Affiliations:** 1Organizing Institution of University Doctoral Studies, Doctoral School of Dentistry, “Titu Maiorescu” University, 67A Gh. Petrascu Street, 040441 Bucharest, Romania; dr.chircaotilia@yahoo.com; 2Faculty of Dental Medicine, “Titu Maiorescu” University, 67A Gh. Petrascu Street, 040441 Bucharest, Romania; corneliabicle@yahoo.com (C.B.); amipangica@yahoo.com (A.M.P.); alexandru.burcea@helpdent.ro (A.B.); 3Department of Mechanics and Strength of Materials, Politehnica University of Timisoara, 1 Mihai Viteazul Avenue, 300222 Timisoara, Romania; 4Department of Materials Science and Physical Metallurgy, Faculty of Material Science and Engineering, University Politehnica of Bucharest, 313 Splaiul Independentei, 060042 Bucharest, Romania; antoniac.iulian@gmail.com

**Keywords:** dental ceramics, adhesive luting systems, FEM simulation, SEM investigation, interface

## Abstract

The purpose of this study is to identify the stress levels that act in inlay and onlay restorations, according to the direction and value of the external force applied. The study was conducted using the Finite Element Method (FEM) of three types of ceramics: pressed lithium disilicate and monolith, zirconia, and three different adhesive systems: self-adhesive, universal, and dual-cure cements. In addition to FEM, the inlay/onlay-dental structure interface analysis was performed by means of Scanning Electron Microscopy (SEM). The geometric models were reconstructed based on computer tomography images of an undamaged molar followed by geometrical procedures of inducing the inlay and onlay reconstructions. The two functional models were then simulated for different orientations of external force and different material properties, according to the considered adhesives and ceramics. The Scanning Electron Microscopy (SEM) was conducted on 30 extracted teeth, divided into three groups according to the adhesive cement type. Both FEM simulation and SEM investigations reveal very good mechanical behavior of the adhesive-dental structure and adhesive-ceramic interfaces for inlay and onlay reconstructions. All results lead to the conclusion that a physiological mastication force applied, regardless of direction, cannot produce a mechanical failure of either inlay or onlay reconstructions. The adhesive bond between the restorations and the dental structure can stabilize the ceramic restorations, resulting in a higher strength to the action of external forces.

## 1. Introduction

Restoring large tooth loss structure of the posterior teeth may be an option, along with direct adhesive technique and indirect adhesive techniques, of which inlay and onlay may be an option because it ensures a safe long-term adaptation with minimally invasive preparation. However, there is controversy about the type of restoration that should be used to restore major defects to avoid fractures and improve survival. The design of an indirect restoration must strike a balance between strictly removing only decayed structures and increasing the strength of the restoration. Removing the marginal ridges and achieving the width and depth of the inlay cavity are the main reasons for the low fracture strength [1]. In the posterior area of the dental arches, the replacement of classical restorations with inlays or onlays is a solution that improves all the parameters on which the success of dental treatment depends [2]. For the restoration of posterior teeth, there are a great variety of materials that correspond aesthetically to the new requirements of patients on the market, such as composite resins, ceramics, and zirconia. Ceramics offer adequate resistance to fractures (160–450 MPa), good survival rates, a high modulus of elasticity, and do not suffer too much abrasion or wear [3]. Ceramic restorations have been introduced in dentistry using traditional feldspar porcelain that is biocompatible and resistant to abrasion and compression forces. Regardless of the method of manufacturing, the inlay and the onlay of glass-ceramic have shown a high survival rate, providing evidence that these restorations are a safe treatment [4]. In recent decades dental ceramics have evolved significantly, especially zirconia, which has proven to be a high-strength dental material [5], but still, problems like fracture and debonding remain the main concerns of restoration failure.

The use of adhesive techniques for cementing ceramic restorations has led to an increase in their mechanical strength. It should be taken into account that proper adhesive cementation can increase the fracture strength of ceramics by up to 69%, which increases the strength and reduces the propagation of cracks [6]. Cementation of ceramic restorations is usually done with self-etch, self-adhesive resins, or dual-cure adhesive resins (light-curing and chemical polymerization). For low-strength feldspar porcelains, the use of the total-etch technique (etch and rinsing) is recommended, followed by the use of dual-cure adhesive cements for crowns and inlays [7]. Self-adhesive cements ensure adhesion by chelating calcium ions by acid groups, producing a chemical bond with the hydroxyapatite of the dental structure. This is a superficial interaction and does not promote the formation of a hybrid layer or smear plug in the dentinal tubules achieved by conventional adhesive cements by the initial application of the adhesive complex with the promotion of the formation of a hybrid layer and a better bond to dentin [8].

Each of these cementation methods has an exact application protocol at the dental level, in one or more stages, which must be strictly observed to obtain the expected results. On the other hand, it is necessary to prepare by etching with 5 and 10% hydrofluoric acid the inner surface of the ceramic to ensure a surface with micro-retentions and increased adhesion [9].

The application of silanes on the etched inner surface of the ceramic ensures good adhesion with the adhesive cement, the silane realizing the connection between the mineral particles of the ceramic and the organic component of the adhesive cement.

One of the challenges produced by cementation is related to the way in which the adhesion at the interfaces of the tooth/adhesive layer/cement/ceramic monolith is achieved. The tighter these areas are, without gaps or fracture lines, the higher the clinical performance.

Aesthetics is not the only criterion that should be discussed in the realization of the restoration treatment plan, but, perhaps more importantly, the chosen materials should be resistant to stress produced by functional forces exerted on the tooth during mastication and also to the action of parafunctional forces that can cause stress on hard dental structures, tissues and periodontal bone. Determining the distribution and analysis of these stresses are of fundamental importance in research and can constantly contribute to reducing the risk of dental restoration failure.

The stress aspects in dental restorations are difficult to determine by experiments, but with an acceptable approximation, the stress-strain state can be evaluated using finite element analysis (FEA). The simulation allows a better understanding of the biomechanics of any dental geometry consisting of a multitude of materials with different mechanical properties [10]. The simulation input usually comes from three directions: geometry, material properties, and loading/fixing conditions. The loading conditions, for example, are coming from real measurements. These reveal that the maximum occlusal force of a healthy man can reach up to 847 N and a healthy woman up to 597 N, while bruxism can increase the maximum occlusal force to over 900 N [11,12,13,14].

Additionally, the elastic properties of natural tooth and restorative materials have to be considered in simulation. Identification of these represents a constant concern in the field of dentistry and material sciences [15,16,17].

The objectives of this study are to identify the values of equivalent stress and strain as well as the deformations that occur according to the direction of applied load on inlay and onlay restorations made of three types of ceramics: IPS e.max CAD-on, Ivoclar Vivadent; ceramics IPS e.max Press, Ivoclar Vivadent; and Novodent GS Zirconia, cemented with three adhesive systems: self-etching Maxcem Elite, Kerr; universal-RelyX Ultimate Clicker, 3M ESPE; and dual-cure Variolink Esthetic LC, Ivoclar Vivadent. In addition, the interface analysis by scanning electron microscopy was conducted, focusing on local aspects at dental structure–adhesive cement interface and also at the interface between the adhesive and the dental ceramic.

## 2. Materials and Methods

### 2.1. Design of Inlay and Onlay Restorations

The construction of the geometric model was accomplished by reconstructing a set of computer tomography images of an unaffected natural molar. This reverse engineering technique allowed us to achieve accurate shape and dimensions of the tooth and to reproduce the finest geometric features [18,19,20]. The reconstruction was made in MIMICS 10.1 (Materialise Inc., Leeuwen, Belgium), in which the 3D geometry was achieved. The refinement and repairing of the structural mesh were then conducted in Geomagic Studio 9 (3D Systems, Morrisville, NC, USA) in order to obtain a valid virtual solid. The rest of the geometric operations were done on SolidWorks 2019 (Dassault Systèmes SE, Vélizy-Villacoublay, France). The starting image collection and the reconstructed molar obtained in this way are presented in Figure 1. The natural reconstructed molar has been truncated with a plan in order to isolate only the crown and cervical area. The purpose of this operation was to reduce the root area, which is not particularly interesting in the analysis, only multiplying the calculation model without bringing an essential effect on the types of preparations to be studied.

Relying on the basic model, two functional models were designed, hereafter referred to as inlay and onlay. These were constructed as two functional assemblies and shaped in such a way that the basic model, the restorations, and the adhesive layer are distinctive parts on which different material properties can be applied. The design stages of the inlay model can be observed in Figure 2. The basic model includes the dentin volume and the enamel shell. On this structure, the inner hole was processed and assembled with two conjugated volumes: one corresponding to the adhesive cement and another corresponding to the restoration geometry. The final model is an assembly containing four distinctive geometric elements.

The onlay model involves performing a reconstruction of the entire upper surface of the dental crown and a transversal cutting, in two planes, for generating the necessary surfaces for virtual restoration. The dimensional jump obtained by profile cutting materializes a particularly important area from a practical point of view: a concentrated stress area both at the level of ceramics and at the level of cement. The stages for onlay model design are presented in Figure 3. As on the inlay model, on the shaped basic model, the adhesive cement part was overlapped, followed by overlapping the ceramic restoration. In this way, the complete onlay model consists of four distinctive geometric elements.

### 2.2. Defining the Materials and Model Association

The simulations were conducted on ANSYS 2019 (ANSYS Inc., Canonsburg, PA, USA) separately for each functional model. For static structural analysis, the following simulation parameters were set: materials corresponding to each structure; contacts between components; loading and fixing conditions; discretization of the structure, and the type of results [21,22,23,24,25].

The commercially available dental materials that were used in the study are presented in Table 1. The elastic properties of restoration structures (longitudinal modulus of elasticity, bending modulus of elasticity, and Poisson’s ratio) were selected from the literature based on experimental trials of either ceramics and cement conducted by manufacturers or independent researchers and are presented in the Table 2 [26,27,28,29,30].

### 2.3. Loading, Fixing Conditions, Contacts and Discretization of the Models

Loading and fixing represent key parameters in simulation. The more accurate these are compared to the natural one, the more realistic simulation becomes. Following this principle, the loading surface was selected on the cuspidal zone for the onlay model and on the transverse groove for the inlay model. A single loading value $\vec{F}$ = 170 N was chosen for both models but applied in six directions, one at a time. The directions simulate the pure shear effect (0°) on the models at one end and the pure compression effect (90°) at the other. The intermediate loadings of 30°, 45°, 60°, 75°, where the combined effect of shear and compression appears in Figure 4. The loading value corresponds to a normal mastication force that may occur at the molar level [11,12,13,14]. The fixed part of both models was at the level of truncation plane, simulating the fixed dental roots in normal conditions. All the displacements of the structure were set free so that the model can deform in any direction.

The bounded contacts definition was done manually between surfaces: dentin-cement, dentin-enamel, inlay/onlay-adhesive, and adhesive-enamel. Ensuring a real contact surface in terms of type, surface dimensions, and position is very important as it directly influences how the stress and strain are transmitted from one element of the model to another.

The discretization of the structure was done in automatic mode using tetrahedral elements of type 10. The continuum structure was divided into discrete elements of the following parameters: the inlay model contains 39,897 nodes and 20,945 elements; the onlay model contains 45,960 nodes and 23,211 elements.

### 2.4. Scanning Light Microscopy Analysis

Verification of the marginal adaptation of ceramic and zirconia inlays to dental structures (dentin and enamel) was performed by an in vitro study. The study was performed on 30 extracted teeth on which inlay and onlay preparations were made and were divided into three groups (Table 3). The preparations were made according to the minimally invasive standard rules, and the cementing was done in strict compliance with the manufacturers’ instructions.

For conducting SEM analysis, the samples were embedded in the resin and sectioned horizontally under cooling, and fixed on a STAB-type aluminum support. The newly formed system was introduced in the Quorum type coating and coated with a 9 nm gold layer to perform the conductivity of the investigated sample at SEM for 60 s. The investigation of the samples was performed using the QUANTA INSPECT F scanning electron microscope equipped with a 1.2 nm resolution field-emission gun (FEG) and energy dispersive X-ray spectrometer (EDS) with the MnK of 133 eV.

## 3. Results

The FEM simulation revealed the stress, strain, and displacement values in all nodes of the discrete structure, in the form of colored maps. In order to have a better interpretation of the results, a series of stress values were extracted from special points of the structure as follows: the stresses in the middle area of the restoration material and the stresses in the middle area of the adhesive cement. The maps showing these stress sampling areas are presented together with the coordinates in the *XoY* plane for the inlay and onlay models in Figure 5. The gray areas represent very low stress values in the structure, while the areas represented in yellow and red are intensely stressed.

The equivalent stresses presented in the paper are the von Mises stress, computed from the principal stresses using Equation (1). The maximum shear stresses presented in the paper are formulated using the principal stresses, according to Equation (2). The equivalent elastic strain is formulated based on Poisson’s ratio and component strain values according to Equation (3).
(1)$$\sigma_{v}=\sqrt{\frac{{\left(\sigma_{1}-\sigma_{2}\right)}^{2}+{\left(\sigma_{2}-\sigma_{3}\right)}^{2}+{\left(\sigma_{3}-\sigma_{1}\right)}^{2}}{2}}$$
(2)$$\tau_{max}=\frac{\sigma_{1}-\sigma_{3}}{2}$$
(3)$$\varepsilon_{eq}=\frac{1}{\sqrt{2}\left(1+\nu \right)}{\left[{\left(\varepsilon_{x}-\varepsilon_{y}\right)}^{2}+{\left(\varepsilon_{y}-\varepsilon_{z}\right)}^{2}+{\left(\varepsilon_{z}-\varepsilon_{x}\right)}^{2}+\frac{3}{2}\left(\gamma_{xy}^{2}+\gamma_{yz}^{2}+\gamma_{xz}^{2}\right)\right]}^{\frac{1}{2}}$$

### 3.1. Inlay Model Simulation

Figure 6 and Figure 7 show the equivalent and shear stress acting in the longitudinal sections of the inlay model. The stress distribution is unbalanced in the case of horizontal loading (0°), and it tends to uniformly distribute in pure compression loading (90°). It is expected that higher cement–ceramic interface stresses to occur for horizontal loading, while more uniform and therefore lower values of stress to be recorded for compression loading at the same interface. The gradual stress distribution can be observed from 0 to 90-degree loading direction, but it appears that the intermediate orientations of 30, 45, and 60 degrees return similar distributions.

The stress values at the control points are given in Table 4 and Table 5, in accordance with the directions of the applied force. The average and standard deviation values are computed on the line and represent the average aspect of the stress in one considered point of the structure. A large standard deviation indicates stress concentration points, where the values significantly change when the loading direction does. On the other hand, low standard deviation shows points in the interface that are more inert to directional changes in loading.

The elastic strain of the structure is shown in Figure 8 and Figure 9 according to the loading direction. This dimensionless parameter depends on the modulus of elasticity of each component and the state of internal loading. The strain represents a very good indicator of how the dental structure reacts in order to deal with the external loadings. Again, symmetric strain in the dentine can be observed for symmetric loadings, while asymmetric strains can be observed for oblique loads. Oblique loadings (15°–45°) produce a significant deformation on one side of the adhesive cement, leaving the other side undeformed.

The variation of stresses according to the sampling point, together with the mechanical strengths of cements and ceramics used in the study, are shown in Figure 10 and Figure 11. Here, the curves for each orientation studied with the mechanical strengths of dental materials found in the literature [28,29,30] were compared. With all stresses and mechanical strength being expressed in MPa, a very good judgment can be made in order to identify possible exceeding of the critical values. Thus, it can be observed that the mechanical strength values of all three ceramics (*cer^I^*, *cer^II^*, and *cer^III^*) exceed by at least an order of magnitude the equivalent stress or shear values recorded at the interface. This confirms that a loading within the physiological mastication values of any direction cannot produce a mechanical failure of the adhesive–ceramic inlay interface. The results can only be reliable in ideal conditions of bonding contact between the two materials. Regarding the cement-biological tissue interface, the stresses in the three adhesive cements (*cem^I^*, *cem^II^, cem^III^*) were highlighted in the same way. It seems that the adhesives’ mechanical strength is much closer to the stress values in the structure but still under the critical point.

### 3.2. Onlay Model Simulation

In the case of the onlay model, the load application surface was in the median area (longitudinal axis) of the molar. The larger contact surfaces between the restoration and the natural dentine-enamel structure allowed a better stress distribution from the reconstruction material to the dentine and the dental roots (fixed surface in our case). In Figure 12 and Figure 13, the equivalent and shear stresses can be observed in the longitudinal section of the model. A stress concentration point can be identified in the vicinity of the dimensional jump of the ceramic restoration. For this reason, the sampling points considered are concentrated in this area. This special site has two weak factors: it has a smaller cross-sectional area than the rest of the restoration and is also very close to the surface where the force is applied. The two factors combined to generate a higher stress state in the model.

The stress values at the sampling points are given in Table 6 and Table 7, corresponding to the directions applied force. Based on these, the stress variation plots were obtained, which were further compared to the mechanical strength of the dental materials, as in the case of the inlay model. The mean and standard deviation represent the average aspect of the stress in the considered sampling points of the interface.

The elastic and shear strains of the structure (Figure 14 and Figure 15) show the preference deformation state of the structure, according to the external loading. The lower elastic modulus materials, dentine in our case, are naturally deformed in order to transmit the internal stresses. In the vicinity of the larger section of the restoration ceramic, the strain value is low (also due to high rigidity material), so the deformations are flowing in a preferred direction, from the dimensional jump of the structure directly to the dentine and eventually to the fixed plane.

Figure 16 and Figure 17 show the stress curves drawn with the considered sampling points, according to the six loading directions, and also the constant curves of mechanical strengths of dental materials and cements. Again, the mechanical strength values of the ceramics used are not exceeded by the shear stresses or equivalent stresses in the structure, even if these curves indicate a significant increase (100–200% for the equivalent stresses) of the stresses in the dimensional jump area. Regarding the cements used in the simulation, it is found that for the onlay model, the maximum stress values are not exceeding the mechanical limit of the materials. This is mainly due to the wider contact surface of the cement in this case, compared to the inlay model.

### 3.3. SEM Analysis of Inlay Model

The attachment of the adhesive cement to the dental structure and to the ceramic inlay was investigated using SEM analysis at 200, 500, and 1000× magnification. According to Figure 18A, there is a very good attachment of cement to the dental structure without cracks, gaps and/or fracture lines. In Figure 18B, the adhesive-inlay cement interface can be observed. It presents an alternation of areas in which the very good attachment of the involved structures and areas with gaps can be observed. According to Figure 18C, there is good adhesion to both the tooth-adhesive system interface and the adhesive-cement-adhesive system and a slight detachment of the inlay from the adhesive cement.

According to Figure 19A, at the border between the two materials, detached micro-fragments from the ceramic are observed, the adhesion being probably lost during the sectioning of the pieces phase. There is no clear demarcation limit of the materials, which suggests that the cement adheres very well to the ceramic. In Figure 19B, there is a perfect adhesion between the adhesive system and the adhesive cement, as well as between it and the dentin determined by the presence of a thin and uniform hybrid layer. According to Figure 19C, the perfect hybridization of the enamel is observed, characterized by the interlocking of the adhesive at the level of the micro retentions created by the action of phosphoric acid.

### 3.4. SEM Analysis of Onlay Model

The SEM investigation on the onlay model was conducted in the same way as for inlay mode. The results can be observed in Figure 20A–C. The adhesive cement layer is homogeneous and well represented and adheres very well to the ceramic component Image (a). Image (b) shows the perfect adhesion of the cement to the ceramic, without gaps and/or fractures, and the presence of silane that penetrated the level of micro retentions obtained by sandblasting and the application of hydrofluoric acid. According to Image (c), there is a detachment of the onlay from the adhesive cement and good adhesion between it and the dentin, with the presence of a dense, narrow, and uniform hybrid layer.

## 4. Discussions

This study examined the behavior of two restoration models, inlay and onlay, subjected to a physiological loading condition of 170 N in value and six directions of orientation. Through the microscopic analysis, the study also examined the way in which the adhesion is made at the level of the restoration/adhesive/tooth interfaces, according to the dental material used. The inlay model simulated a frusto-conical ceramic inlay located in the central area of the molar and inserted into the bone structure to a depth of 4.2 mm. Based on the simulated structure, the equivalent stresses, shear stresses, equivalent strains, and shear strain were extracted from the surface and inside of the model in accordance with the properties of ceramic and adhesive dental materials. The onlay dental reconstruction model was designed to simulate a situation as unfavorable as possible from the mechanical point of view. This was achieved by a dimensional jump at the level of the ceramic element, in the vicinity of which a high concentration of stresses is expected.

A very recent study that analyzed the transmission of stresses in ceramic inlays showed that the highest values of stress occurred in teeth restored with inlay, the stresses recorded on onlay being lower. The authors found the highest stresses on the zirconium inlay. The lowest stresses were observed in the dental structures, cement, and at the interface between the restoration and the dental tissue, according to our study [17].

For onlay, restorations made of lithium disilicate ceramic offered the highest breaking strength, especially when the restorations were cemented with dual-setting resin, but the differences were of no statistical significance. Therefore, the fracture strength of onlay is not significantly influenced by the material used to manufacture them [3,31].

In our study, the shear stress distribution at the outer level of the molar showed a gradual increase with the transition from 0° to 90° loading direction. Equivalent stresses, on the other hand, varied significantly only between the two extreme loading directions: vertical (0°) and horizontal (90°). It was also shown that 170 N loading-value could not produce mechanical fracturing effects of the ceramic inlay manufactured from the considered materials regardless of which direction it was oriented. Related to the effect of cavity design on stress distribution, onlay design protected dental structures more effectively than inlay models, according to other researchers [1,31].

According to another FEA study performed on CAD-CAM ceramic restorations (IPS e.max CAD-on) and loaded on the occlusal and occlusal-vestibular surface, no significant differences were reported in the cement layer or between different preparation designs [32]. Özkir SE conducted his FEA study to determine the stress distributed on the tooth and onlay-type restorations made of integral ceramic and composite resin, reporting that the highest stress concentration was observed at the ceramic restoration (3.77 GPa) while the lowest value of stress was recorded on the tooth (1.69 GPa) [33]. Stress concentration sites indicate the likely onset of tooth failure and fracture. High elastic modulus materials (porcelain, for example) tend to accumulate high stresses but fail to transfer those stresses further to the tooth structure, and therefore avoid crown fractures [33,34].

Another recent study that analyzed the strength of occlusal and occlusal-vestibular restorations to the action of a force of 300 N showed that teeth restored with IPS e.max CAD presented the highest stress at the level of restoration, the lowest stress of the substrate teeth, and the lowest probability of failure for the overall system, without reporting significant differences in the cement layer or between different preparation designs [33,34].

Inlay restorations have a higher fracture strength when made of lithium disilicate ceramic. Onlays made of lithium disilicate ceramic offered the highest breaking strength, especially when the restorations were cemented with dual-setting resin [3,31].

Cementation seems to be a critical step, and its long-term success is based on adherence to clinical protocols. Proper management of adhesive cementation protocols requires knowledge of adhesive principles and adherence to the clinical protocol to achieve a lasting connection between the tooth structure and restorative materials. The adhesive bond between the restorations and the dental structure can stabilize the ceramic restorations, resulting in higher resistance to the action of external forces. The values of mechanical strength of adhesives being much lower than the mechanical strength of ceramics lead to the focus on adhesive interface behavior. Variolink Esthetic dual-cure cement has a mechanical strength limit very close to the stress values obtained in the critical points of the structure. This could cause a structural failure in inlay restorations, especially when the load is applied application oblique or tangent. In the onlay model, mainly due to the wider contact surface of the cement, the maximum elastic limit is not exceeded. However, for an accidental increase of loading value over the simulated 170 N, the Maxcem self-adhesive cement properties can be easily exceeded. Cementation of restorations based on lithium disilicate ceramic used in this study is done after etching the inner surface with hydrofluoric acid, which aims to increase the adhesion by infiltrating the adhesive cement in the micro retentions obtained from this action. In addition, the silanization process increases the ceramic–cement adhesion by the connection between the ceramic silica and the organic cement matrix [35].

SEM images of the self-adhesive cement-inlay interface reveal areas with very good attachment interrupted by gaps and, in the case of the onlay, a homogeneous, well-represented layer of adhesive cement that adheres very well to the ceramic. When evaluating the universal adhesive cement, detached ceramic micro fragments can be seen at the inlay-cement interface, perfect adhesion to the ceramic, and the presence of the silane penetrating the micro-retentions in the case of onlay. When evaluating the adhesion of dual-cure cement (Variolink Esthetic), a separation of the interfaces is observed for both zirconia inlay and onlay. The SEM analysis of the adhesive cement-dental structure interface exposes a very good adhesion of self-adhesive cement (Maxcem) without cracks, gaps, and/or fracture lines. There is also a good adhesion of the dual-cure cement (Variolink Esthetic), highlighted by the presence of a dense, narrow, and uniform hybrid layer. In the case of universal cement (RelyX Ultimate), there is the perfect hybridization of the enamel and the presence of a thin and uniform hybrid layer in the dentin area; this aspect is observed by Aguiar et al., which highlighted the formation of a uniform hybrid layer of high density and long resin smear plugs [36].

## 5. Conclusions

The FEA simulation and SEM investigation upon the inlay and onlay restoration structures lead to several synthetic conclusions:The mechanical strength values of the ceramics (*cer^I^, cer^II^, cer^III^*) are superior by at least one order of magnitude to the equivalent and shear stress values in either inlay and onlay models. This confirms that a real loading value similar to a simulated one cannot produce a mechanical failure of any of the ceramics.The adhesive cements, especially dual-cure *cem^III^*, had a mechanical strength limit very close to the stress values obtained in the critical points of the structures. This could lead in practice to fracture initiation sites, especially when the applied load is oblique or tangent to the molar.Shear stresses show jumps of 100–200% due to the dimensional leap of onlay reconstruction. This type of structure is at least geometrically inferior to the inlay, making it a good candidate for crack failure when accidental force occurs.The adhesion between the restorations and the tooth structure can stabilize the ceramic restorations, resulting in higher resistance to the action of external forces.The adhesive cement/restoration interface seems to be more difficult to achieve in inlay; self-adhesive and universal cements seem to be more efficient in onlay type restorations.The adhesive cement/dental structure interface is much more efficient both for the types of design and for all types of cementing techniques.

## Figures and Tables

**Figure 1 materials-14-05048-f001:**
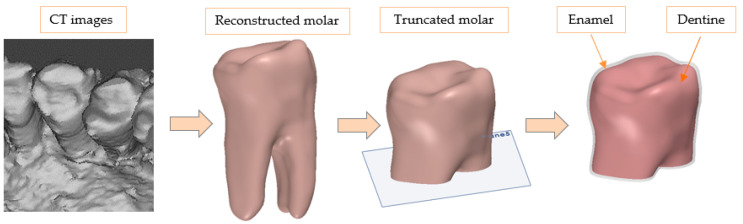
Stages of obtaining the basic geometric model.

**Figure 2 materials-14-05048-f002:**
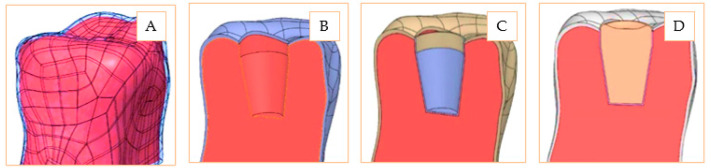
Design stages of inlay model: Basic model including dentine and enamel volumes (**A**); Inner surface generation (**B**); Adhesive cement layer applied to the model (**C**); Restoration ceramic insertion (**D**).

**Figure 3 materials-14-05048-f003:**
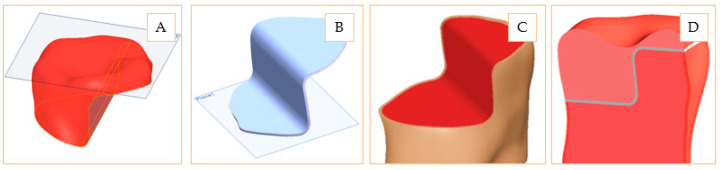
Design stages of onlay model: Ceramic restoration (**A**); Adhesive cement (**B**); Dentin and enamel model (**C**); Complete onlay model (**D**).

**Figure 4 materials-14-05048-f004:**
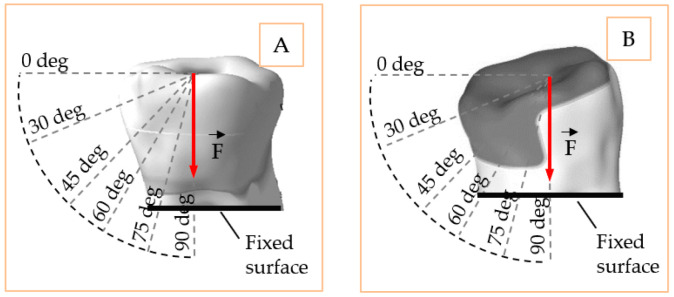
Loading and restraining conditions of the inlay model (**A**) and onlay model (**B**).

**Figure 5 materials-14-05048-f005:**
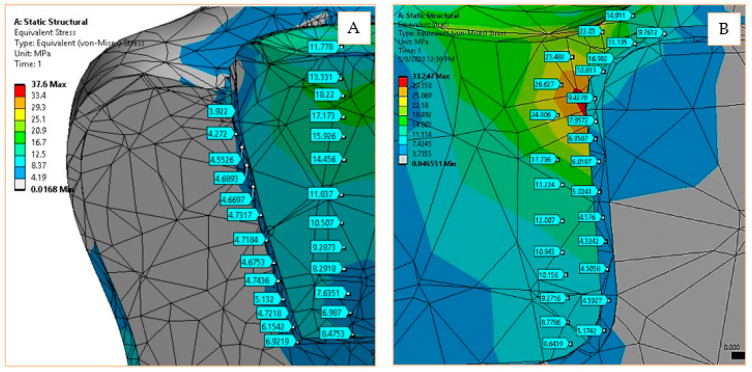
Stress sampling points of inlay model (**A**) and onlay model (**B**).

**Figure 6 materials-14-05048-f006:**
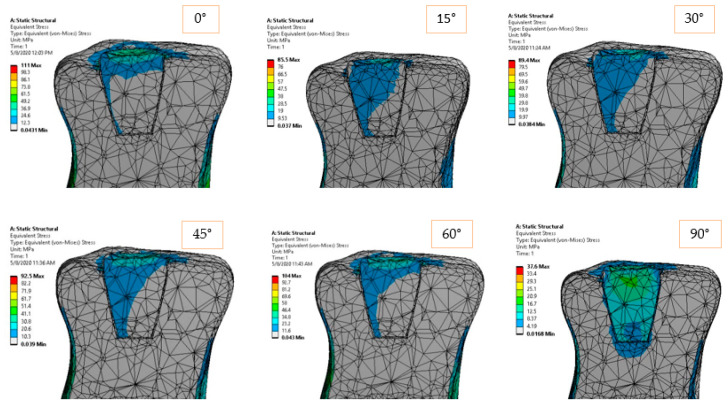
Equivalent elastic stress for 170 N load and 0°–90° loading direction.

**Figure 7 materials-14-05048-f007:**
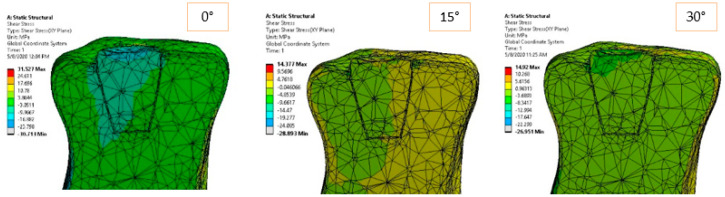
Shear stress in *XY* plane for 170 N load and 0°–90° loading direction.

**Figure 8 materials-14-05048-f008:**
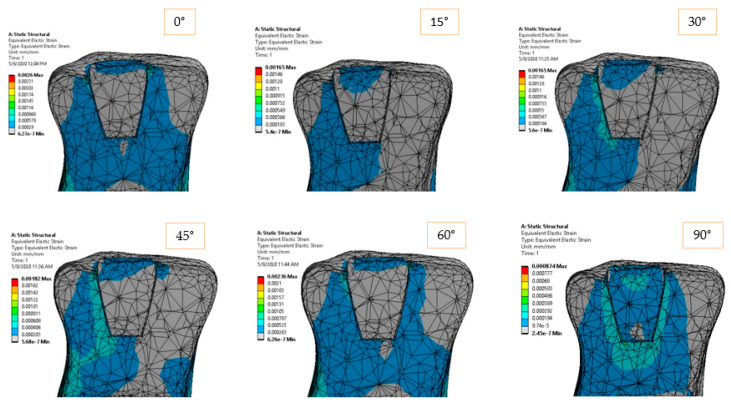
Equivalent elastic strain for 170 N load and 0°–90° loading direction.

**Figure 9 materials-14-05048-f009:**
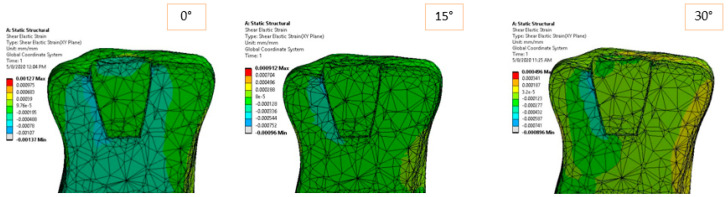
Shear strain in *XY* plane for 170 N load and 0°–90° loading direction.

**Figure 10 materials-14-05048-f010:**
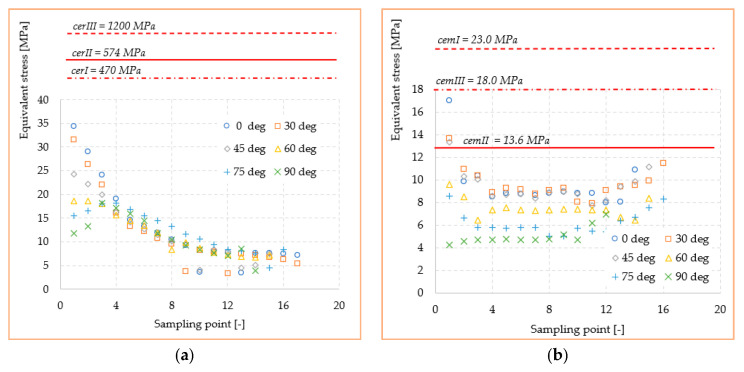
Mechanical strength of dental materials and equivalent stresses recorded at interface: (**a**) ceramic; (**b**) adhesive cement.

**Figure 11 materials-14-05048-f011:**
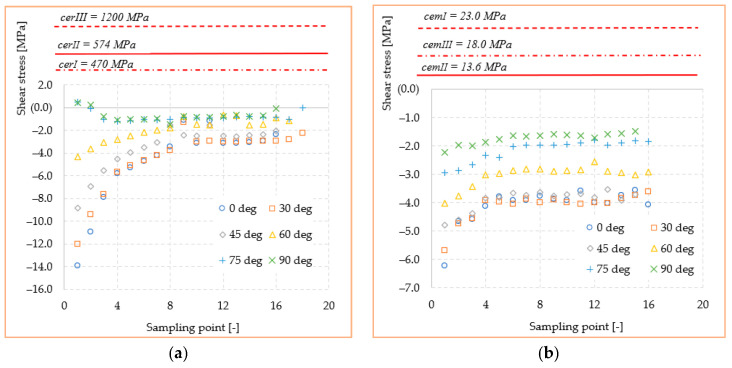
Mechanical strength of dental materials and shear stresses recorded at interface: (**a**) ceramic; (**b**) adhesive cement.

**Figure 12 materials-14-05048-f012:**
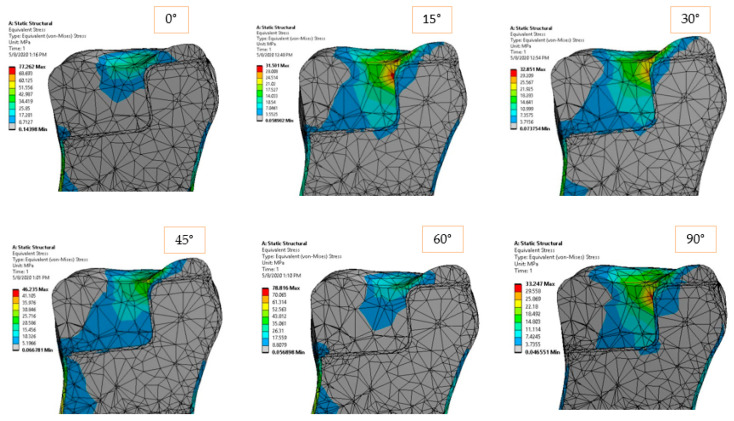
Equivalent elastic stress for 170 N load and 0°–90° loading direction.

**Figure 13 materials-14-05048-f013:**
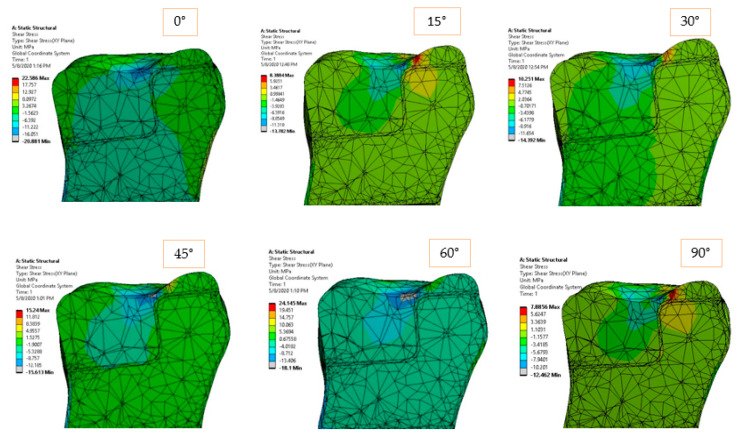
Shear stress in *XY* plane for 170 N load and 0°–90° loading direction.

**Figure 14 materials-14-05048-f014:**
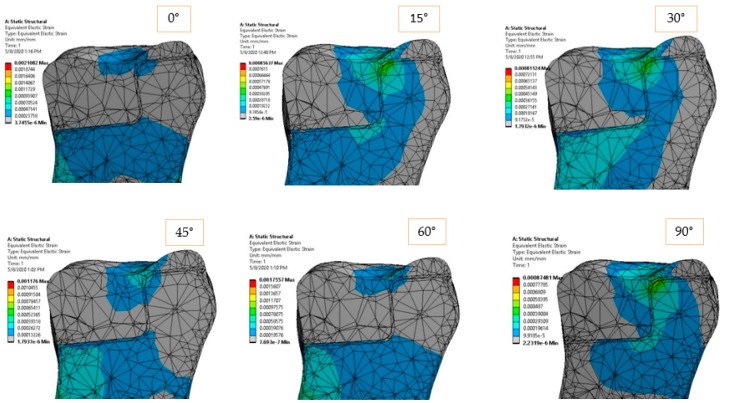
Equivalent elastic strain for 170 N load and 0°–90° loading direction.

**Figure 15 materials-14-05048-f015:**
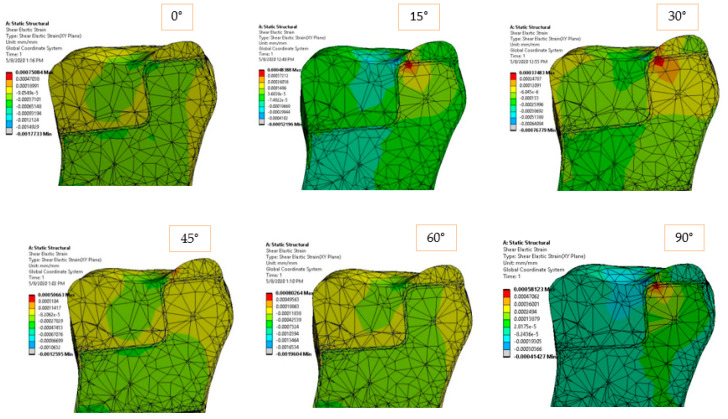
Shear strain for 170 N load and 0°–90° loading direction.

**Figure 16 materials-14-05048-f016:**
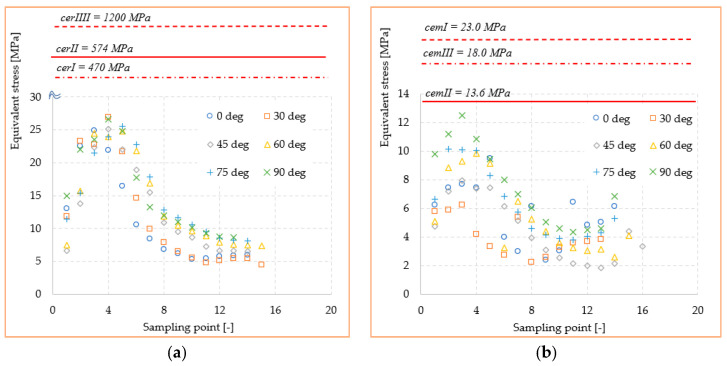
Mechanical strength of dental materials and equivalent stresses recorded at interface: (**a**) ceramic; (**b**) adhesive cement.

**Figure 17 materials-14-05048-f017:**
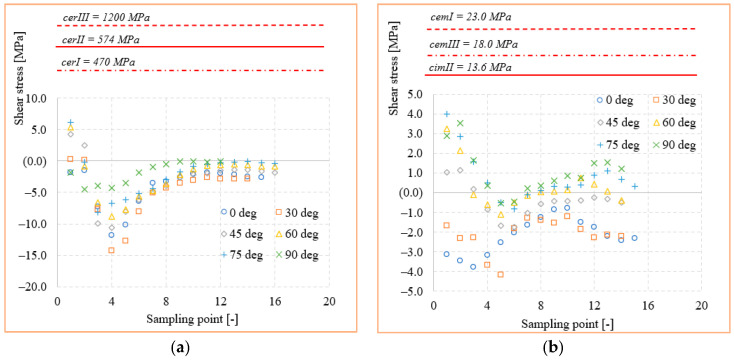
Mechanical strength of dental materials and shear stresses recorded at interface: (**a**) ceramic; (**b**) adhesive cement.

**Figure 18 materials-14-05048-f018:**
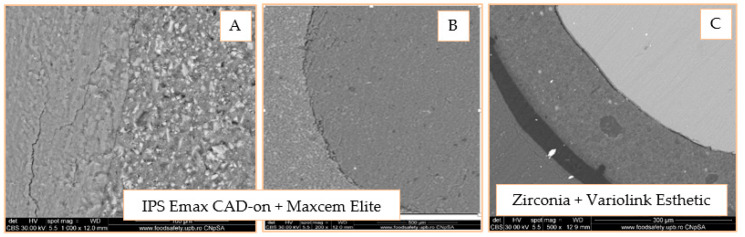
SEM 1000× image of the tooth area—adhesive cement (**A**); SEM 200× image of the IPS e.max adhesive cement-inlay area (**B**); SEM image, 500× of the enamel—adhesive system—adhesive cement—–inlay (**C**).

**Figure 19 materials-14-05048-f019:**
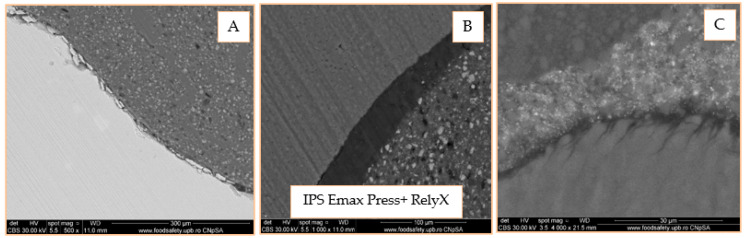
Adhesive coating interface, 500× (**A**); SEM 1000× image of the tooth area (dentin)—adhesive—adhesive cement (**B**); Adhesive-adhesive-enamel cement area, 4000× (**C**).

**Figure 20 materials-14-05048-f020:**
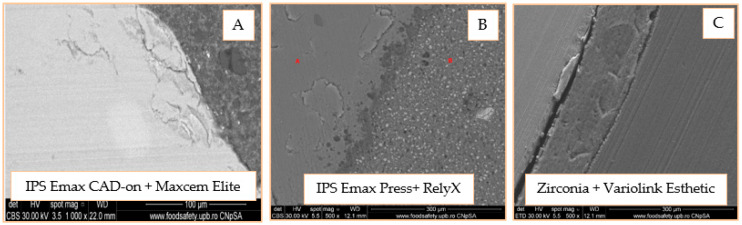
SEM 1000× image of the onlay area—adhesive cement (**A**); SEM 500× image of the onlay area—adhesive cement (**B**); SEM 500× image of the onlay area—adhesive cement–dentin (**C**).

**Table 1 materials-14-05048-t001:** Restoration type and associated commercially available materials.

Cement Type	Code	Ceramic Type	Code	Restoration
Universal RelyX Ultimate Clicker	*cem^I^*	Pressed IPS e.max Press, Ivoclar Vivadent	*cer^I^*	inlay, onlay
Self-adhesive Maxcem	*cem^II^*	Carbonat IPS e.max CAD-on, Ivoclar Vivadent	*cer^II^*	inlay, onlay
Duble cure Variolink	*cem^III^*	Zirconium oxide Zirconia Novodent GS	*cer^III^*	inlay, onlay

**Table 2 materials-14-05048-t002:** Mechanical properties of considered dental materials.

Material	Structure	Density [kg/m^3^]	Young Modulus/ Bending Modulus [MPa]	Poisson’s Ratio [-]
*cer^I^*	inlay/onlay restoration	2480	82,300	0.22
*cer^II^*	inlay/onlay restoration	2480	82,300	0.23
*cer^III^*	inlay/onlay restoration	2480	88,000	0.34
*cem^I^*	adhesive cement	2400	7700	0.24
*cem^II^*	adhesive cement	2350	4400	0.24
*cem^III^*	adhesive cement	2400	8100	0.25
*dentine*	dentine	2000	17,000	0.30
*enamel*	enamel	2750	74,000	0.23

**Table 3 materials-14-05048-t003:** Distribution of tooth groups for SEM analysis.

Material	Group 1	Group 2	Group 3
*Restaurative material*	IPS e.max CAD-on Ivoclar Vivadent	IPS e.max Press Ivoclar Vivadent	Novodent GS Zirconia
*Adhesive cement*	Maxcem Elite Kerr	RelyX Ultimate 3M ESPE	Variolink Esthetic

**Table 4 materials-14-05048-t004:** Stress values in restorative ceramic.

No.	Loading Direction						Average Stress [MPa]	Standard Deviation [MPa]
	0°	30°	45°	60°	75°	90°		
1	34.37	31.53	24.34	18.60	15.45	11.78	22.68	9.01
2	29.08	26.28	22.17	18.60	16.56	13.33	21.00	5.98
3	24.18	22.02	20.01	18.02	17.96	18.22	20.07	2.56
4	19.00	16.03	16.31	15.62	18.19	17.17	17.05	1.32
5	14.67	13.23	14.19	14.43	16.82	15.93	14.88	1.29
6	13.33	12.17	12.68	13.36	15.54	14.46	13.59	1.23
7	11.96	10.78	11.55	11.89	14.44	11.84	12.08	1.24
8	10.48	9.51	10.20	8.30	13.32	10.51	10.39	1.66
9	9.23	3.72	9.60	9.90	11.63	9.29	8.89	2.68
10	3.55	8.20	4.02	8.70	10.66	8.29	7.24	2.82
11	7.90	7.91	8.44	7.70	9.33	7.64	8.15	0.64
12	7.73	3.35	7.97	7.29	8.36	6.99	6.95	1.83
13	3.42	7.45	4.42	6.91	8.10	8.48	6.46	2.06
14	7.61	7.19	5.02	6.78	7.55	3.92	6.34	1.52
15	7.58	6.78	7.48	7.12	4.45	6.92	6.72	1.16

**Table 5 materials-14-05048-t005:** Stress values in adhesive cement.

No.	Loading Direction						Average Stress [MPa]	Standard Deviation [MPa]
	0°	30°	45°	60°	75°	90°		
1	17.04	13.69	13.35	9.58	8.54	4.27	11.08	4.53
2	9.83	10.98	10.29	8.53	6.60	4.55	8.47	2.46
3	10.38	10.40	10.04	6.43	5.77	4.69	7.95	2.61
4	8.49	8.90	8.54	7.34	5.81	4.67	7.29	1.71
5	8.82	9.26	8.69	7.54	5.71	4.73	7.46	1.85
6	8.78	9.15	8.76	7.32	5.77	4.72	7.42	1.83
7	8.63	8.76	8.40	7.28	5.82	4.68	7.26	1.68
8	8.84	9.10	8.86	7.36	5.03	4.74	7.32	1.99
9	8.97	9.25	8.95	7.40	5.03	5.13	7.45	1.95
10	8.80	8.06	8.79	7.39	5.69	4.72	7.24	1.69
11	8.81	7.91	7.75	7.31	5.45	6.15	7.23	1.23
12	7.96	9.10	8.16	7.32	5.40	6.92	7.48	1.27
13	8.03	9.38	9.39	6.69	6.37	6.93	7.80	1.35
14	10.92	9.51	9.88	6.42	6.67	6.94	8.39	1.94
15	9.68	9.90	11.18	8.40	7.54	6.94	8.94	1.60

**Table 6 materials-14-05048-t006:** Stress values in restoration ceramic.

No.	Loading Direction						Average Stress [MPa]	Standard Deviation [MPa]
	0°	30°	45°	60°	75°	90°		
1	12.98	11.89	6.64	7.41	11.42	14.91	10.88	3.22
2	22.54	23.34	13.78	15.74	15.40	22.03	18.81	4.27
3	24.94	22.76	22.32	24.47	21.50	23.48	23.24	1.31
4	21.92	26.94	25.17	23.92	23.99	26.63	24.76	1.89
5	16.48	21.68	22.00	24.80	25.51	24.81	22.55	3.37
6	10.51	14.66	18.91	21.79	22.77	17.74	17.73	4.58
7	8.44	9.87	15.44	16.91	17.89	13.22	13.63	3.83
8	6.82	7.86	10.92	11.88	12.84	12.01	10.39	2.46
9	6.16	6.53	9.48	10.41	11.64	10.94	9.20	2.32
10	5.36	5.48	8.64	9.61	10.61	10.16	8.31	2.33
11	5.40	4.75	7.30	8.80	9.52	9.27	7.51	2.04
12	5.71	5.14	6.56	7.89	8.58	8.78	7.11	1.53
13	5.84	5.43	6.56	7.61	8.23	8.64	7.05	1.31
14	5.96	5.47	6.47	7.42	8.10	8.56	7.00	1.23
15	6.01	4.42	6.46	7.35	7.98	8.26	6.75	1.43

**Table 7 materials-14-05048-t007:** Stress values in adhesive cement.

No.	Loading Direction						Average Stress [MPa]	Standard Deviation [MPa]
	0°	30°	45°	60°	75°	90°		
1	6.22	5.77	4.74	5.07	6.63	9.76	6.37	1.80
2	7.42	5.89	7.19	8.85	10.15	11.20	8.45	1.99
3	7.67	6.23	7.92	9.26	10.10	12.50	8.94	2.20
4	7.42	4.18	7.37	9.84	10.03	10.81	8.27	2.46
5	9.47	3.36	7.44	9.11	8.26	9.43	7.84	2.33
6	3.97	2.75	6.12	3.23	6.83	7.96	5.14	2.12
7	2.98	5.37	5.11	6.47	5.74	6.96	5.44	1.39
8	6.14	2.25	3.96	5.25	4.60	6.02	4.70	1.46
9	2.40	2.61	3.07	4.40	4.16	5.02	3.61	1.07
10	3.05	3.30	2.54	3.59	3.86	4.58	3.49	0.70
11	6.44	3.60	2.16	3.25	3.78	4.32	3.93	1.43
12	4.84	3.67	1.98	3.03	4.01	4.51	3.67	1.04
13	5.02	3.82	1.81	3.15	4.28	4.59	3.78	1.16
14	6.11	3.79	2.12	2.59	5.28	6.85	4.46	1.92
15	6.03	3.66	4.41	4.09	4.99	5.88	4.84	0.96

## Data Availability

The data presented in this study are available on request from the corresponding author.

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
