# Peer review of "Adhesive-Ceramic Interface Behavior in Dental Restorations. FEM Study and SEM Investigation"

_materials, 2021, doi:10.3390/ma14175048_

Round 1

Reviewer 1 Report

This interesting study provides useful findings in terms of the dental material science, however, few result and discussion are provided to support the FEM results from a material science point of view. I believe that this manuscript are very interesting for either dental clinicians and researchers in dental materials science, and I think it would be appropriate to submit the paper to a journal focusing on dental materials.

Due to the small size and low resolution of the figures, the details of the experimental results were not clear.

Reviewer 2 Report

Article is not writen according to Journal standard adn is to long.

Introduction provides sufficient background and include all relevant references. Reader gets a detail overview of the presented topic.

Research design is appropriate. Numerical simualtion is supported with real results. Methods are clearly described and persented.

Results are clearly presented in figures and tables. In the figures numbers are not clearly visible to reader.

Figure 2 & 3 --> can be smaller, so that all 4 figures (a, b, c and d) come in one row.

Figure 5 & 6 -->  Frame of the figure is visible. In the figures numbers are not clearly visible to reader.

Figure 7 -23 -->  In the figures numbers are not clearly visible to reader.

Tables are not made according to journal standard.

Conclusion is supported by results.

Reviewer 3 Report

The main aim of the paper was to characterize the deformations that occur according to the direction of stress applied on inlay and onlay restorations made of 3 types of ceramics cemented with three adhesive systems. The topic is up-to-date, methods well-chosen, results well described and discussion extensive. However, there are some weak points that should be improved before publication:

  1. Descriptions in some figures are completely illegible (font too small) – look especially at figures 8, 10, 11, 12, 13, 14, 15, 18, 19, 20, 22, 23
  2. there are a few abbreviations that should be developed when they appear in the text for the first time (e.g. CT scan)
  3. Why authors chose the load 170 N
  4. The novelty of research is not described enough – please highlight this aspect of research
  5. There are some language mistakes: e.g.: “However, this time too cim2 (Maxcem) could have the properties exceeded by an accidental increase of the load over the 170 N simulated” – please replace the word too with also
